# Deciphering Structural Determinants in Chondroitin Sulfate Binding to FGF-2: Paving the Way to Enhanced Predictability of Their Biological Functions

**DOI:** 10.3390/polym13020313

**Published:** 2021-01-19

**Authors:** Giulia Vessella, José Antonio Vázquez, Jesús Valcárcel, Laura Lagartera, Dianélis T. Monterrey, Agatha Bastida, Eduardo García-Junceda, Emiliano Bedini, Alfonso Fernández-Mayoralas, Julia Revuelta

**Affiliations:** 1Department of Chemical Sciences, University of Naples Federico II, Via Cinthia 4, I-80126 Naples, Italy; giulia.vessella@unina.it (G.V.); ebedini@unina.it (E.B.); 2Group of Recycling and Valorization of Waste Materials (REVAL), Marine Research Institute (IIM-CSIC), Eduardo Cabello, 6, 36208 Vigo, Spain; jvazquez@iim.csic.es (J.A.V.); jvalcarcel@iim.csic.es (J.V.); 3Institute of Medicinal Chemistry (CSIC), Juan de la Cierva 3, 28006 Madrid, Spain; l.lagatera@iqm.csic.es; 4BioGlycoChem Group, Institute of General Organic Chemistry (CSIC), Juan de la Cierva 3, 28006 Madrid, Spain; d.monterrey@iqog.csic.es (D.T.M.); agatha.bastida@csic.es (A.B.); eduardo.junceda@csic.es (E.G.-J.); alfonso.mayoralas@csic.es (A.F.-M.)

**Keywords:** glycosaminoglycan, chondroitin sulfate, structure-activity relationships, protein interactions, fibroblast growth factor 2

## Abstract

Controlling chondroitin sulfates (CSs) biological functions to exploit their interesting potential biomedical applications requires a comprehensive understanding of how the specific sulfate distribution along the polysaccharide backbone can impact in their biological activities, a still challenging issue. To this aim, herein, we have applied an “holistic approach” recently developed by us to look globally how a specific sulfate distribution within CS disaccharide epitopes can direct the binding of these polysaccharides to growth factors. To do this, we have analyzed several polysaccharides of marine origin and semi-synthetic polysaccharides, the latter to isolate the structure-activity relationships of their rare, and even unnatural, sulfated disaccharide epitopes. SPR studies revealed that all the tested polysaccharides bind to FGF-2 (with exception of CS-8, CS-12 and CS-13) according to a model in which the CSs first form a weak complex with the protein, which is followed by maturation to tight binding with *k*_D_ ranging affinities from ~1.31 μM to 130 μM for the first step and from ~3.88 μM to 1.8 nM for the second one. These binding capacities are, interestingly, related with the surface charge of the 3D-structure that is modulated by the particular sulfate distribution within the disaccharide repeating-units.

## 1. Introduction

Chondroitin sulfate (CS) is a family of polysaccharides ubiquitously distributed in the extracellular matrix (ECM) and on cell surfaces usually found attached to proteins in the form of proteoglycans. These are linear polysaccharides composed of alternating 4)-β-GlcA-(1→3)-β-GalNAc-)1→disaccharide units (GlcA = glucuronic acid, GalNAc = *N*-acetylgalactosamine) whose hydroxyl groups can be sulfated in variable positions and quantities, giving rise to different forms of CSs (Figure 1a) [1]. Chondroitin sulfates extracted from terrestrial animal sources are hybrid structures which bear mainly 4-*O*- or 6-*O*-GalNAc monosulfated disaccharides (CS-A and CS-C respectively), and a low percentage of non-sulfated disaccharide (CS-0). However, disulfated (CS-B, CS-D, CS-E, CS-K and CS-L), and even trisulfated disaccharides (CS-M, CS-S and CS-T) have been found in marine species (Figure 1b) [2].

CSs interact with various signaling molecules (growth factors (GFs), receptors and guidance molecules), and such interaction plays a key role in the control of numerous biological processes (cell migration, growth, differentiation, guidance and development) and in diverse pathologies (metastasis, neurodegeneration and inflammation) [3]. Central to these interactions is the sulfation group distribution along the CS backbone, which encodes specific functional information, namely key molecular recognition elements of the specific sulfation motifs [4,5,6].

However, the polydispersity and heterogeneity of natural CSs, in combination with non-template driven biosynthesis, and the difficulty of accessing defined structures have been major barriers to understanding the molecular basis of such interactions. To overcome this, recent efforts have been centered on the bottom-up synthesis of defined tailored sequences (an expensive and time-consuming strategy) and subsequent evaluation in order to elucidate the effect of structural parameters (length, sulfation pattern, etc.) on their recognition by signaling molecules [7,8]. In addition to being labor intensive, this approach is generally limited to the most abundant sulfate profiles (CS-C, CS-A and CS-E), and it lacks rare sulfated epitopes that can encode relevant biological information, hampering detailed structure-activity relationship investigations and subsequent biomedical applications [9]. To solve this drawback, during the last decade a “semi-synthetic” alternative approach has emerged [10] which has not only afforded CS-species with a well-defined sulfation distribution [1,11], but also new polysaccharides with unprecedented sulfation profiles [12,13].

On the other hand, given the large number of binding partners and diverse GAG structures, new approaches need to be developed that enable us to look more rapidly and more globally across protein families and GAG classes [14]. Very recently, we have proposed a new “holistic approach” to look globally at how particular sulfate distribution within the disaccharide epitope directs the binding of chondroitin sulfates to growth factors. Specifically, we have observed that this disposition modulates the surface charge of the polymers’ 3D-structure, which interestingly, has a significant influence on their capacity to binding to growth factors and, consequently, on their functional effects [15]. Inspired by these results, we have envisioned that a thorough understanding of the physical-chemical properties of CSs and how these relate to their protein binding will lead to enhanced predictability of their biological functions (Figure 2).

This proposal raises of the question whether CSs can be tailored on the basis of their disaccharide epitopes in such a way that they are able to stimulate desired functions while not inducing unwanted ones. However, controlling CS activity in order to exploit its immense potential ultimately requires a more comprehensive understanding of how the particular sulfate distribution within the disaccharide epitope can impact on their biological activities. 

In this study, we systematically analyzed the structure-activity relationships (SARs) of several CSs, isolating the effect of the different sulfate-bearing motifs. To do so, a binding affinity screening was conducted using surface plasmon resonance (SPR) and the zeta potential as the physical-chemical indicative parameter, in accordance with our approach [15]. As example of a target-protein we selected fibroblast growth factor 2 (FGF-2 or basic fibroblast growth factor). Like other FGF family members, FGF-2 is involved in a large number of biological processes in a GAG cofactor-assisted manner [16,17], and a rising amount of evidence has shown that CSs participate and modulate its binding to its FGFR receptor [18].

## 2. Materials and Methods

### 2.1. Materials

CS-6 was purchased from Creative Biomart and was dialysed against deionized water before use. For disaccharide compositional analysis, Chondroitinase ABC was purchased from Sigma-Aldrich (EC 4.2.2.4., 1.66 U mg^−1^, product number C2905, Sigma-Aldrich, Saint Louis, MO, USA) and unsaturated disaccharide standards were obtained from Grampenz (Aberdeen, UK). Commercial-grade reagents and solvents were used without further purification, except where differently indicated. The term “deionized water” refers to water purified by a Millipore Milli-Q gradient system. Dialyses were conducted on Specra/Por 3.5 kDa cut-off membranes (Spectrum Laboratories, Inc., Rancho Dominguez, CA, USA) at 4 °C. Freeze-drying was performed with a 5Pascal Lio 5P 4K freeze dryer (5Pascal, S.r.l., Milan, Italy). Centrifugations were performed with an Eppendorf centrifuge 5804R instrument (Eppendorf AG, Hamburg, Germany) at 4 °C (3500 rpm, 5 min).

### 2.2. Production of CSs from Marine Sources 

Highly purified chondroitin sulfates (CSs) were produced using cartilage wastes from *Chimaera monstrosa* (CS-1), *Galeus melastomus* (CS-2), *Prionace glauca* (CS-3), *Scyliorhinus canicula* (CS-4), and *Raja clavata* (CS-5), through reported procedures [19,20,21,22]. The disaccharide composition was determined by strong anion exchange (SAX) chromatography after enzymatic digestion with chondroitinase ABC from *Proteus vulgaris* at 0.2 U mg^−1^ of CS, in accordance with a previously described procedure [21]. Briefly, a solution of natural polysaccharides (0.05 M Tris-HCl/0.15 M sodium acetate buffer, pH 8, 37 °C) was treated with the enzyme for 24 h, after which it was inactivated by heating at 70 °C for 25 min and separated by centrifugation (12,857G). Supernatants were finally filtered (0.2-µm polyethersulfone (PES) syringe filters) and analyzed based on a previously reported method [23]. An external calibration curve was built with each standard to calculate the amount of disaccharide units in the sample and was reported as a percentage of the weight.

### 2.3. Preparation of Semi-Synthetic CSs

#### 2.3.1. Semi-Synthesis of CS-7

CS-7 was prepared by sulfation of a known polysaccharide **1** (a semi-synthetic partially protected chondroitin polysaccharide prepared as reported in [24]), according to a previously reported sulfation method [25] (Figure 3). Briefly, polysaccharide **1** (42.4 mg, 88.8 µmol) was dissolved in dry dimethylformamide (DMF) (1.4 mL) and then treated with a 1.2 M solution of SO_3_·py complex in dry DMF (1.51 mL). After overnight stirring at 50 °C, a saturated NaCl solution in acetone (13 mL) was added at room temperature; the obtained precipitate was collected by centrifugation, then suspended in deionized water (2 mL). The mixture (pH ≈ 2) was heated to 50 °C and, after 1.5 h of stirring, was treated at room temperature with a 4 M NaOH(aq) to adjust the pH to 12. After overnight stirring at room temperature, 1 M HCl(aq) was added until neutralization. Dialysis and subsequent freeze-drying produced polysaccharide CS-7 (46.3 mg, 109% weight yield).

#### 2.3.2. Semi-Synthesis of CS-9 and CS-10

These polysaccharides were prepared from CS-6 according to two procedures described previously by us [15]. These procedures are described in the Appendix A.

#### 2.3.3. Semi-Synthesis of CS-13, CS-14 and CS-15

The polysaccharides were prepared in accordance with multi-step procedures developed previously by us [13]. To do this, *Escherichia coli* O5:K4:H4 sourced chondroitin [26] was employed as the starting material. The synthetic procedures are described in Appendix A.

#### 2.3.4. Semi-Synthesis of CS-8, CS-11 and CS-12

These polysaccharides were obtained from CS-15, CS-14 and CS-13 respectively according to a previously reported procedure of mild sulfation procedure [27] (Figure 4). CS-15 (or CS-14 or CS-13) (38.2 mg, 75.9 µmol) was dissolved in deionized water (1.5 mL) and passed through a short DOWEX 50 WX8 column (H^+^ form, 20–50 mesh, approx. 4 cm^3^). Elution with deionized water was continued until pH of the eluate was neutral. The eluate was then treated with some drops of aqueous TBAOH (16% *w/v* TBAOH in water) to neutralize the solution. Freeze-drying of the collected eluate resulted in product 2 (or 3 or 4), which was then dissolved in dry DMF (1.5 mL). The obtained solution was cooled to 4 °C and treated with a 0.42 M solution of SO_3_·py complex in dry DMF (575 µL). After 1 h of stirring at 4 °C, some drops of 1 M NaHCO_3_(aq) were added at room temperature to adjust the pH to 7; 0.3 M NaCl(aq) (7.3 mL) was then added at room temperature and the stirring was continued for 1 h. The solution was dialyzed for one day and subsequently passed through a short DOWEX 50 WX8 column (H^+^ form, 20–50 mesh, approx. 4 cm^3^). Elution with deionized water was continued until the pH of the eluate was neutral. The eluate was then treated with some drops of 1 M NaOH(aq) to adjust the pH to 7. Dialysis and subsequent freeze-drying produced polysaccharide CS-8 (or CS-11 or CS-12) (37.1 mg, 97% weight yield).

### 2.4. Characterization of Sulfated Polysaccharides

#### 2.4.1. Disaccharide Composition

The disaccharide composition of natural CSs was analyzed by chondroitinase ABC digestion of the polysaccharides into disaccharide subunits, followed by separation using an Agilent 1260 HPLC (Agilent Technologies, Inc., Santa Clara, CA, USA).

#### 2.4.2. Nuclear Magnetic Resonance

NMR spectra were recorded on a Bruker DRX-600 (Bruker Corp., Billerica, MA, USA) (operating at 600 and 150 MHz for ^1^H and ^13^C, respectively), on a Varian INOVA-500 (Varian Medical Systems, Inc., Palo Alto, CA, USA) (operating at 500.13 and 125.76 MHz for ^1^H and ^13^C, respectively), all equipped with cryo-probe, and on a Varian MERCURY 400 (Varian Medical Systems, Inc., Palo Alta, CA, USA) (operating at 400 and 100 MHz for ^1^H and ^13^C, respectively). Samples were dissolved in D_2_O, and acetone (^1^H: (C*H*_3_)_2_CO at δ 2.22 ppm; ^13^C: (*C*H_3_)_2_CO at δ 31.5 ppm) or DMSO-*d_6_* (^1^H: C*H*D_2_SOCD_3_ at δ 2.49 ppm; ^13^C: *C*D_3_SO*C*D_3_ at δ 39.5 ppm) were employed as the internal standard. For gradient-selected COSY and TOCSY experiments we used spectral widths of either 6000 Hz in both dimensions and data sets of 2048 × 256 points. TOCSY experiments were recorded using a mixing time of 120 ms. HSQC-DEPT experiments were recorded using the ^1^H-detected mode via single quantum coherence with proton decoupling the 13C domain. We used data sets of 2048 × 512 points, with typically 60 increments. Finally, in HSQC-TOCSY and HMBC we employed data sets of 2048 × 256 points, with 120 increments, and the mixing time for HSQC-TOCSY was set to 120 ms.

#### 2.4.3. Molecular Weight Determination

The number and average molecular weights were determined using size exclusion chromatography (SEC) (Agilent Technologies, Inc., Santa Clara, CA, USA) according with analytical methods recently extensively described [21,28]. The eluent was 1% acetic acid in distilled water (pH  =  3.7). The SEC was conducted at room temperature.

#### 2.4.4. Zeta Potential Analysis

Zeta potential measurements were performed on a Malvern Zetasizer Nano ZS (Malvern Instruments, Herrenberg, Germany). Samples (1 mg/mL) were dissolved in an aqueous solution of NaCl (0.01 mM) and each experiment was carried out in triplicate.

#### 2.4.5. Circular Dichroism

Circular dichroism (CD) spectra were acquired using a Jasco-815 dichrograph (JASCO International Co. Ltd., Tokyo, Japan), previously calibrated with D-10-camphorsulphonic acid. The path length of the cuvette (Hellma GmbH & Co. KG, Müllheim, Germany) was 1 cm. Total polysaccharide concentration in the cuvette was 1 mg/mL in buffer (HEPES 2.5 mM, pH 7.4) for all samples. UV-CD spectra were recorded from 200 to 240 nm with step size of 1 nm and bandwidth of 1 nm at 25 °C. 

### 2.5. Determination of Binding Affinity of Polysaccharides with FGF-2 Using Surface Plasmon Resonance (SPR)

The surface of a CM5 sensor chip (Biacore Inc., GE Healthcare, Boston, MA, USA) was activated using freshly mixed *N*-hydroxysuccimide (NHS; 100 mM) and 1-(3-(dimethylamino)propyl)-ethylcarbodiimide (EDC; 400 mM) (1/1, *v/v*) in water. Next, FGF2 (50 μg/mL) in aqueous NaOAc (10 mM, pH 5.0) was passed over the surface until a ligand density of 7.000 RUs. Quenching of the remaining active esters was accomplished by passing aqueous ethanolamine (1.0 M, pH 8.5) over the surface of the chip. The control flow cell was activated with NHS and EDC, which was then treated with ethanolamine. HBS-EP (0.01 M HEPES, 150 mM NaCl, 3 mM EDTA, 0.05% polysorbate 20; pH 7.4) was employed as the running buffer for immobilization, binding and affinity analysis. Serial dilutions of each compound in HBS-EP buffer at a flow rate of 30 μL/min was employed for association and dissociation at a temperature of 25 °C. One 30 s injections of aqueous NaCl (2.0 M) at flow rate of 30 μL/min was employed for regeneration to achieve baseline status. Affinity data were fitted, except for CS-13, to a two sites affinity model, yielding two *k*_D_’s (for an example see Appendix A). The competition experiments were prepared with samples containing a fixed concentration of CS-7 (10 µM) and a series of concentrations of the polysaccharide CS-15 which range from 1 to 50 µM in HBS-EP. The regeneration conditions were similar to the binding experiments described above. The evaluation was made using a BIAcore X100 evaluation software (Biacore Inc., GE Healthcare, Boston, MA, USA).

## 3. Results and Discussion

To validate our proposal (see Figure 2), herein, we conducted a binding-affinity analysis of a library of CSs, with the FGF-2 growth factor. To systematize the analysis, initially, we selected five natural CSs (CS-1, CS-2, CS-3, CS-4, and CS-5) that differ from one another in one of their disaccharide topological motifs. Additionally, CS-6 (an LMWCS of animal origin) and four CSs were semi-synthetized (CS-7, CS-8, CS-9 and CS-10) to recapitulate individually the effect of these motifs (Figure 5). We also carried out a comprehensive analysis of a library of semi-synthetic CSs carrying sulfate groups placed at C-2 and/or C-3 positions of the GlcA residues, alone or in combination with the C-6 sulfation of the GalNAc residues (CS-11, CS-12, CS-13, CS-14 and CS-15) (Figure 5) [10].

The aim of this analysis was to allow us to complete the panel of disaccharide epitopes with very rarely found motifs of marine species and non-natural ones with respect to their recognition by target proteins. It is important to note that although actual knowledge is scarce, recent studies have demonstrated that CS possessing K subunits with 3-*O*-sulfonated GlcA units displays a neurite outgrowth activity comparable to natural CS-E species [29].

### 3.1. Characterization of Polysaccharides

The disaccharide composition of natural CSs was analyzed by chondroitinase ABC digestion of the polysaccharides into disaccharide subunits, followed by separation using high-performance liquid chromatography (HPLC) [23,30] (Table 1).

In respect of the semi-synthetic polysaccharides, a 2-D nuclear magnetic resonance (NMR) spectroscopy investigation was performed. The ratio between different kind of sulfated residues was measured by integration of DEPT-HSQC spectra, assuming that the signals to be compared displayed similar ^1^*J*_CH_ coupling constants and that the difference of around 5–8 Hz from the experimental set value did not cause a substantial variation in the integrated peak volumes [31]. For CS-9, CS-10, CS-13, CS-14, and CS-15, this integration was accomplished by comparison with literature data [13,15]. The ratio between GalNAc(6S,4S) and GalNAc units in CS-7 was estimated to be 80:20 by integration of the GalNAc *O*-6 methylene signals (δ_H/C_ 4.24/69.1 and 3.77/62.5), corresponding to disulfated and non-sulfated GalNAc residues, respectively (Figure 6a). For CS-8 and CS-11, the ratio between GalNAc(6S) and GalNAc units was calculated by integration of the signals corresponding to 6-*O*-sulfated and non-sulfated GalNAc residues (see Figure 6b,c). Finally, for CS-12, the ratio between GalNAc(6S) and GalNAc units was calculated by integration of the signals corresponding to CH-5 signals of GalNAc(6S) and GalNAc residues (see Figure 6d). For full NMR assignments of these polysaccharides, see Appendix A.

Table 2 summarizes the disaccharide composition and sulfation degree of semi-synthetic CSs and CS-6.

The structural characterization of polysaccharides was completed with determination of their molecular mass using high-performance size exclusion chromatography (see Appendix A).

### 3.2. Influence of Polysaccharide 3D-Sulfate Distribution on Their Binding Capacities with FGF-2

#### 3.2.1. Analysis of Natural Chondroitin Sulfates

To determine the influence of CS sulfate motifs on the binding to FGF-2, five marine-source CS variants were chosen. These had molecular weights in the range of 45-70 kDa. Three polysaccharides had sulfate groups at positions (6S), (4S) and (2S,6S) in different proportions (CS-3-5), and the other two also had lower amounts of other sulfate motifs ((4S,6S) and (2S,4S)) in variable proportions (CS-1-2) (Table 1). The later were chosen to profile the possible effect of minority sulfate distributions on binding preferences to FGF-2.

Based on our proposal (see Figure 2), we firstly examined the degree of sulfation dependence and the effect of substituting CS sulfo groups in the arrangement of sulfates on the surface. With this aim, zeta potential (a physico-chemical parameter that is indicative of the polysaccharide surface charge) was measured, and circular dichroism (DC) spectra were recorded. As can be observed in Figure 7a, the surface charge seems to be independent of the sulfation degree. It is worth noting that for polysaccharide CS-4 an unexpected decrease in the zeta potential was observed in spite of its similar degree of sulfation (DS) to CS-3.

Recently, we have proposed that 6-sulfated disaccharides favor arrangement of sulfates on the surface, while 4-sulfate ones have a negative effect [15]. In this point, it is important to note that, in our previous study, we have analyzed only polysaccharides with 6S- and 4S-sulfated residues. However, in the marine polysaccharides analyzed herein, other sulfated residues could have important effects in Z potential values and these must be considered to stablish relation-ships. A clear example of these possible effects is the important decrease of zeta potential observed for polysaccharide CS-4 (−41.7 mV). In this case, and according to our previous results, the enrichment in 4S should involve an important increase in zeta-potential value due to the negative effect of this these disaccharides in the arrangement of sulfates on surface. This unexpected result could enable profiling of an unknown effect of minor sulfated-disaccharides along the CS backbone in the arrangement of sulfates on the helical 3-D structure of polysaccharides. Indeed, the CD spectrum showed the negative band at 210 nm that has previously been described for polysaccharides rich in α-helical content [32], including those with rare sulfate distributions (Figure 7b).

To validate our hypothesis that this arrangement directs the binding of CSs to GFs, SPR (surface plasmon resonance) experiments were carried out. Gratifyingly, the SPR response units for the various compounds correlated well with the surface charge, even for CS-4, being indicative of the important role that minor sulfated-disaccharide epitopes play not only in the arrangement of sulfates on the surface, but also in the binding process to GFs. Subsequently, titration experiments were performed with a selected set of natural CSs (CS-1, CS-2 and CS-4). The titration curves (for an example, see Appendix A) for all polysaccharide fitted well to a two-state binding mode, and *k*_D_ values were determined (Table 3). A model in which the CSs first form a weak complex with the protein, which is followed by maturation to tight binding, can rationalize these results [33]. The shapes of sensograms of CS-1, CS-2 and CS-4-FGF-2 interactions seem to be similar and to have a comparable *k*_D_, suggesting the importance of the minor sulfated-disaccharide epitopes in the binding to GFs.

#### 3.2.2. Analysis of Semi-Synthetic Chondroitin Sulfates with Defined Sulfation Profile

With these results in mind, and to gain insight into the role of disaccharide epitopes, we analyzed a library of polysaccharides with a defined sulfation profile, isolating different disaccharide epitopes. In this case, to mimic the structure of possible domains, polysaccharide M_w_ was in the range of 7–10 kDa. To do a systematical analysis we analyzed three CS types; with sulfo groups (one or two) located at GalNAc residue (CS-6, CS-7, CS-9, and CS-10), at GlcA residue (CS-13, CS-14, and CS-15), and at both GalNAc and GlcA residues (CS-8, CS-11, and CS-12).

When GalNAc was modified, the data were similar to that observed previously by us [15] (for a comparison between this data with previously obtained see Appendix A)Thus, 6-*O*-sulfated moieties (CS-10) again seem to induce a disposition of the sulfate groups pointing outside the 3D-helical structure, while the presence of 4-*O*-sulfated ones (CS-9) (even with a small proportion of 6-*O*-sulfated subunits (CS-6)) only induces a disposition of the sulfate groups inside the helix, as can be deduced by their Z-potential values (−21.3 vs. −16.8 mV) (Figure 8a,b). These differences had consequences for their recognition by FGF-2, demonstrating that this growth factor has preference for CS-10 in comparison to polysaccharides CS-9 and CS-6 (Figure 8c).

On the other hand, compounds having sulfate groups at GlcA residue showed very different behavior, depending on the sulfation pattern. The zeta potential value observed for CS-15 (−42.8 mV) suggest that 2-*O*-sulfated motifs particularly favor arrangement of sulfate charges on the helical surface. This behavior was further supported by comparing the zeta potential value of polysaccharides CS-13 (−31.0 mV), an isomer having a sulfate ester at C-3 position of the same ring (Figure 8a). As expected, these zeta-potential value variations resulted in important differences in binding affinity to FGF-2. Thus, much higher affinity was observed for 2-*O*-sulfated chondroitin sulfate (CS-15) than for the 3-*O*-sulfated one (CS-13) (Table 4, entries 3 and 4). In fact, for the last polysaccharide, the only *k*_D_ observed (6.6 µM) (Table 4, entry 4) suggests that CS-13 forms a weak, non-specific complex with the protein, not-showing the subsequent tight binding. As can be observed from CS-14, this negative effect of C-3 sulfation at GlcA moiety can be countered in part by sulfation of C-2 position in the same ring, which further supports the relevance of the 2-sulfated position. In this case, besides the slight decrease in zeta-potential value, it is important to note the important effect that 2-*O*-sulfation produces in 3-D structure, increasing the α-helical content to similar levels to those of CS-15. This structural behavior seems to be the origin of the large increase in binding affinity observed for CS-14 (*k*_D1_ = 8.6 µM, *k*_D2_ = 33.7 nM) (Table 4, entry 5). A similar trend was observed with CS-11, a polysaccharide obtained by partial sulfation (~50%) at the C-6 position of GalNAc moieties of CS-14. Strikingly, the binding affinity of this polysaccharide to FGF-2 (*k*_D1_ = 1.33 µM, *k*_D2_ = 4.75 nM) (Table 4, entry 8) was even higher than CS-14, although a slight increase in zeta potential was observed (−31.0 vs. −35.2 mV), corroborating again the importance of molecular architecture in these binding processes [32].

Conversely, when CS-13 and CS-15 were sulfated (~20%) at the C-6 position of GalNAc to afford CS-12 and CS-8, respectively, a different effect was observed. In both cases, SPR results indicated that these polysaccharides display distinct binding preferences that the rest of polysaccharides, adjusting to a model in which a strong binding of FGF-2 to the polysaccharide occurs first, followed by a weak interaction with the protein. These results could be in consonance with the previously observed effects that 6-*O*-sulfated GalNAc residues have on the binding capacity and specificity of chondroitin sulfates binding to growth factors [34], although more studies are needed to elucidate the mechanism of action.

Last, but not least, an interesting effect was observed when CS-7 (a disulfated polysaccharide that possesses both sulfated motifs in the GalNAc ring) was analyzed. In this case, it is surprising to note that despite the total loss of α-helix structural motifs due to the absence of the band at 210 nm in CD spectrum (Figure 8b), this polysaccharide showed a good responsiveness when interaction with FGF-2 was measured (*k*_D1_ = 1.31 μM, *k*_D2_ = 3.88 nM) (Table 4, entry 2). This result was very surprising given that chondroitin sulfates explored thus far exhibited a clear helical structure. Many previous reports have shown the importance of recognizing this molecular architecture in terms of the defined periodic spacing between sulfate moieties and the helical layout of sulfate moieties [35,36]. In fact, previous results have showed that spinal cord CS (composed mainly by 4S- and 6-S monosulfated residues) binds FGF-2 with greater affinity that the more highly sulfated CS-E of natural origin (composed by a mixture of different disaccharides with over 50% E-units), which is indicative that the arrangement of disaccharides is critical to their biological function [37,38].

To gain an insight into this unexpected effect, an SPR solution/surface competition assay was performed to examine the effect of helical structure on the FGF-2-polysaccharide interaction. To do this, CS-7 and CS-15 were selected, two polysaccharides that produced a similar responsiveness against FGF-2 (see entries 2 and 3 of Table 4) even though they had completely different 3-D structures (Figure 8b). As can be observed in Figure 9a for FGF-2-CS-7 interaction, no competition effect was detected when different concentrations of CS-15 were added. In contrast, binding signals increased substantially when CS-15 was present in the CS-7 solution, and an increase in concentration of the first polysaccharide resulted in a greater responsiveness.

This suggests that FGF-2 recognizes simultaneously CS-7 and CS-15, indicating that it seems to tolerate derivatives having or not 3-D defined structures. Nevertheless, and based on these and our previous results [39,40], we can hypothesize that the helical layout of sulfate moieties would allow a productive binding mode. This could be related with the biding between growth factor–growth factor receptor and chondroitin sulfate, who would act as co-receptor, leading to enhanced signal transduction across the plasma membrane. On the contrary, non-structured polysaccharides would sequester the protein, protecting the growth factor from proteolysis or locally blocking its active site (Figure 10). Nevertheless, in order to demonstrate this hypothesis, a deep insight into the influence of the 3D-helical structure in the selectivity of the binding to a specific region of the protein as well as its consequence in biological function is required.

Much interesting work lies ahead in our efforts to further develop this paradigm. While we have explored and investigated a limited number of chondroitin sulfate sequences in this study, generation of a significant diversity of sequences will be useful to reveal and exploit the full extent of selectivity that chondroitin sulfates can offer. Work is in progress to this aim and will be published as soon as possible elsewhere.

## 4. Conclusions

In this work, we have analyzed different natural chondroitin sulfates of marine origin, as well as semi-synthetic ones, isolating the effects of their rare sulfate-bearing motifs in the binding to FGF-2. Our results highlight that the particular sulfate distribution within the disaccharide repeating-units plays a key role in the binding of FGF-2, modulating the surface charge of the 3D-structure that, according with our hypothesis, has a significant influence on the binding capacity. Furthermore, our data revealed that rare sulfated epitopes and even unnatural ones may have a similar, if not better, effect in the binding affinity to growth factors that CS-6 and -4 isomers, the most abundant disaccharides in natural derivatives. Finally, we have observed that although sulfation profiles provide the primary basis for modulating interactions with growth factors, the helical layout of sulfate charges or not could be determinative in terms of the specificity of the zone-binding, which could have important biological consequences.

## Figures and Tables

**Figure 1 polymers-13-00313-f001:**
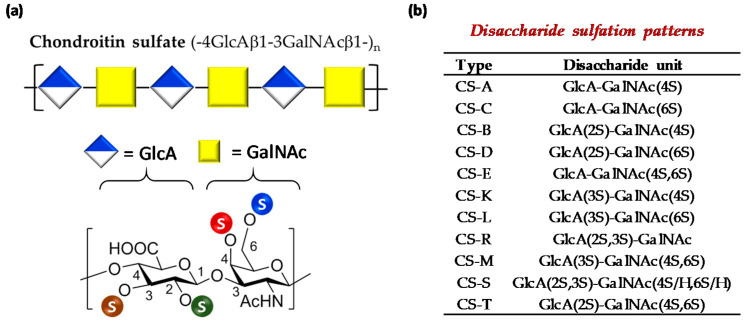
(**a**) Structure of chondroitin sulfate chains (top); chemical structure of typical disaccharide units found in CS chains (bottom). An “S” enclosed by circles indicates the various positions that can be esterified by sulfates. (**b**) Disaccharide sulfation patterns of natural CSs.

**Figure 2 polymers-13-00313-f002:**
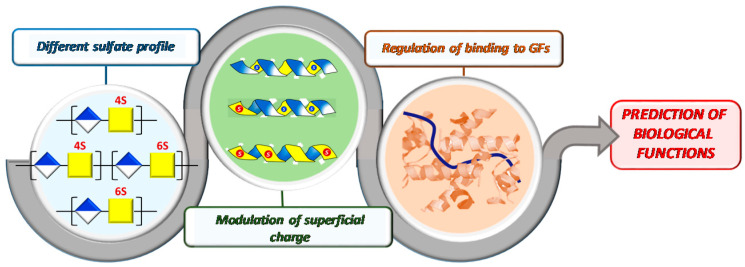
Schematic diagram of our proposal.

**Figure 3 polymers-13-00313-f003:**
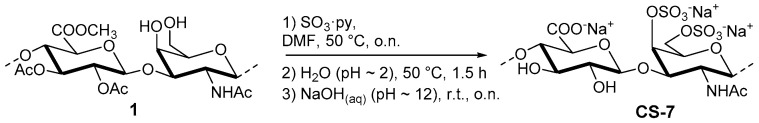
Scheme of the semi-synthesis of CS-7.

**Figure 4 polymers-13-00313-f004:**
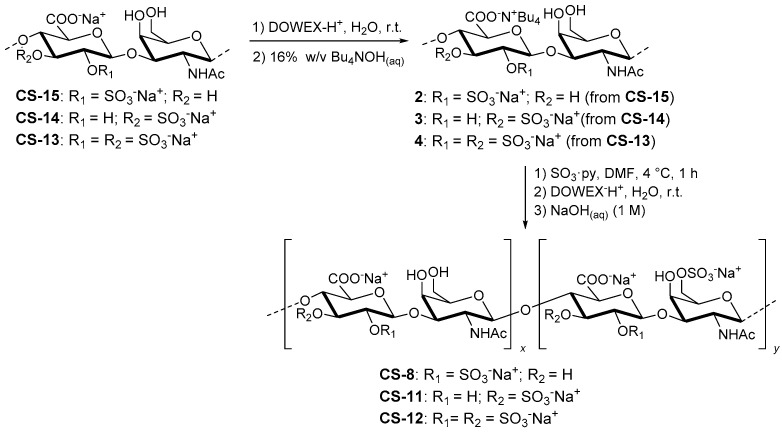
Scheme of the synthesis of CS-8, CS-11 and CS-12.

**Figure 5 polymers-13-00313-f005:**
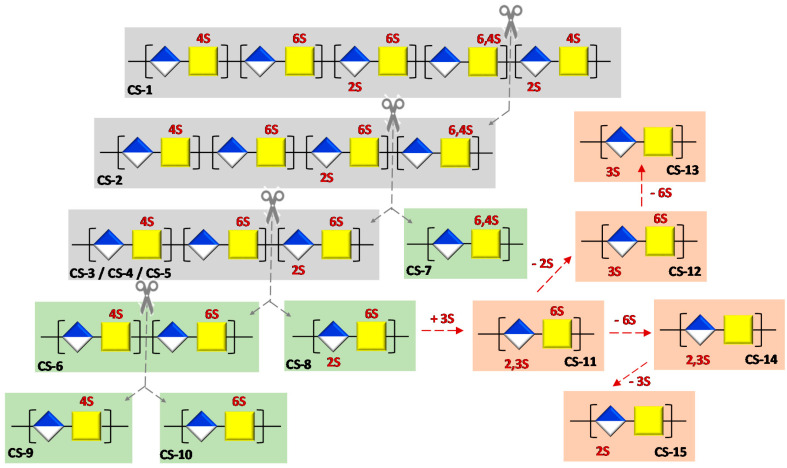
Schematic representation of the polysaccharides systematically analyzed in this study. Natural CSs are highlighted with a grey background. The grey dashed lines indicate differences in terms of disaccharide motifs among natural CSs. These motifs were isolated in the polysaccharides that are highlighted with a green background. Finally, a red background indicates polysaccharides found to recapitulate rare or unnatural sulfation patterns in GlcA. The differences between these polysaccharides are shown over the red dashed arrows.

**Figure 6 polymers-13-00313-f006:**
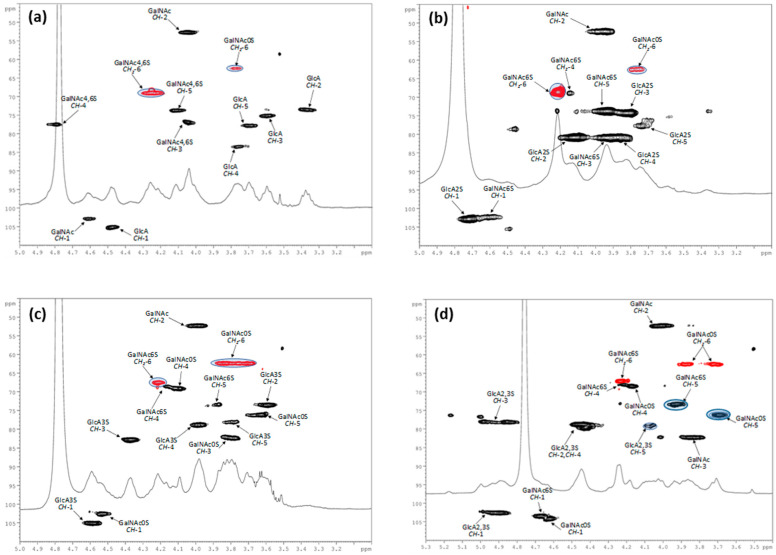
^1^H and DEPT-HSQC NMR spectra (400 MHz, D_2_O, 298 K, zoom) of CS-7 (**a**), CS-8 (**b**), CS-11 (**c**), and CS-12 (**d**). In Figure 6a, GalNAc is referred to both 4,6-disulfated and zero-sulfated GalNAc units. In Figure 6b–d, GalNAc are referred to both 6-sulfated and zero-sulfated GalNAc units. Densities enclosed in the highlighted areas were integrated for disaccharide residue ratio estimation. For full ^1^H and DEPT-HSQC NMR spectra, see Appendix A.

**Figure 7 polymers-13-00313-f007:**
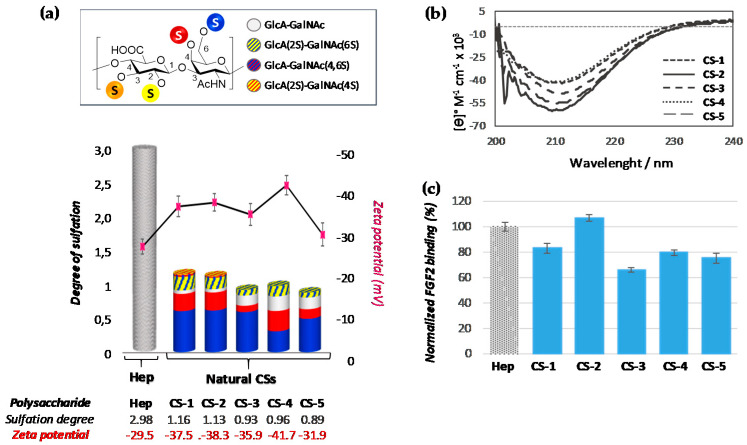
(**a**) Degree of sulfation of polysaccharides (their numerical values are indicated at the bottom in grey). The pink points indicate the zeta-potential values (their numerical values are indicated at the bottom, in red). The sulfated-disaccharide composition of each polysaccharide is shown in colours (for interpretation, see figure legend). (**b**) CD spectra of polysaccharides. (**c**) Bar graphs of normalized FGF-2 binding. The concentrations of polysaccharides were 0.5 mg/mL. All bar graphs are based on triplicate experiments. Heparine (Hep) was employed as reference.

**Figure 8 polymers-13-00313-f008:**
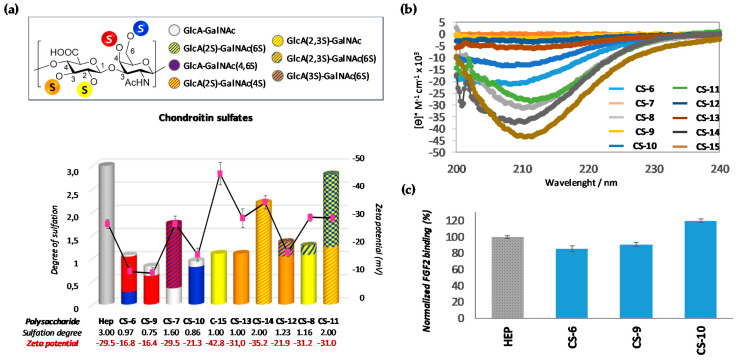
(**a**) Degree of sulfation of polysaccharides (their numerical values are indicated at the bottom, in grey). The pink points indicate the zeta-potential values (their numerical values are indicated at the bottom, in red). Sulfated-disaccharide composition of each polysaccharide is shown in colors (for interpretation, see figure legend). (**b**) CD spectra of polysaccharides. (**c**) SPR binding of immobilized FGF-2 to GalNAc sulfated derivatives (CS-6, CS-9 and CS-10) (0.5 mg/mL) normalized against heparin. All bar graphs are based on triplicate experiments.

**Figure 9 polymers-13-00313-f009:**
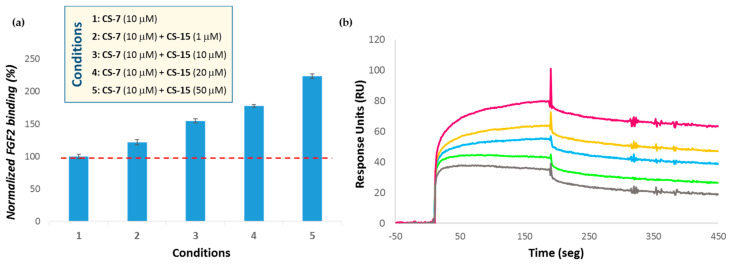
(**a**) Bar graph of normalized CS-7 binding preference to surface FGF-2 by competing with different concentrations of CS-15. The dashed red line indicates the lever of responsiveness of CS-7 alone. (**b**) SPR sensograms for binding-affinity measurements. The concentration of CS-7 was, in all cases, 10 µM, and concentrations of CS-15 were (from top to bottom): 50 µM, 20 µM, 10 µM and 0 µM.

**Figure 10 polymers-13-00313-f010:**
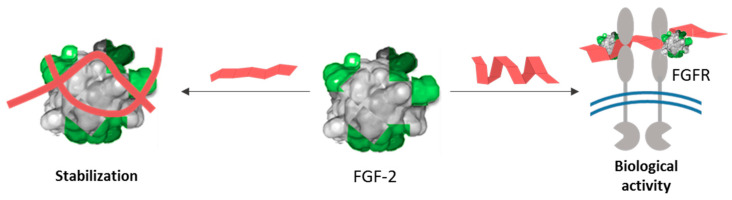
Dual function of chondroitin sulfates in their binding to FGF-2 as mediators of growth factor signaling to cells (**right**) or as providers of storage sites, protecting and stabilization the protein (**left**).

**Table 1 polymers-13-00313-t001:** Disaccharide composition and corresponding sulfation pattern of natural polysaccharides. Sulfation degree: average number of sulfate residues per disaccharide unit.

	*C. monstrosa* (CS-1)	*G. melastomus* (CS-2)	*P. glauca* (CS-3)	*S. canicula* (CS-4)	*R. clavata* (CS-5)
GlcA-GalNAc(4S)^[a]^	22.1 ± 0.3	23.8 ± 0.1	10.1 ± 0.1	31.5 ± 0.6	16.0 ± 0.1
GlcA-GalNAc(6S)^[a]^	52.7 ± 0.1	54.9 ± 0.4	64.2 ± 0.4	32.4 ± 0.1	55.9 ± 0.1
GlcA-GalNAc^[a]^	4.4 ± 0.2	4.2 ± 0.5	16.3 ± 0.6	22.5 ± 1.4	19.5 ± 0.1
GlcA(2S)-GalNAc(6S)^[a]^	17.4 ± 0.1	15.0 ± 0.1	9.3 ± 0.2	16.3 ± 0.2	8.6 ± 0.1
GlcA-GalNAc(4,6S)^[a]^	2.4 ± 0.1	1.5 ± 0.0	n.o^[b]^	n.o^[b]^	n.o^[b]^
GlcA(2S)-GalNAc(4S)^[a]^	1.0 ± 0.0	0.6	n.o^[b]^	n.o^[b]^	n.o^[b]^
Sulfation degree	1.16	1.13	0.93	0.96	0.89
4S/6S ratio	0.32	0.34	0.15	0.64	0.26

[a] Results expressed as mean % ± standard deviation (n = 2); [b] n.o.: not observed.

**Table 2 polymers-13-00313-t002:** Disaccharide composition and corresponding sulfation pattern of semi-synthetic polysaccharides.

Polysaccharide	Disaccharide Units	Ratio	Sulfation Degree
CS-6^[a]^	GlcA-GalNAc(6S)/GlcA-GalNAc(4S)/GlcA-GalNAc	25:72:3	0.97
CS-7	GlcA-GalNAc(4,6S)/GlcA-GalNAc	80:20	1.60
CS-8	GlcA(2S)-GalNAc(6S)/GlcA(2S)-GalNAc	16:84	1.16
CS-9	GlcA-GalNAc(4S)/GlcA-GalNAc	75:25	0.75
CS-10	GlcA-GalNAc(6S)/GlcA-GalNAc	86:14	0.86
CS-11	GlcA(2S,3S)-GalNAc/GlcA(2,3S)-GalNAc(6S)	44:56	2.56
CS-12	GlcA(3S)-GalNAc/GlcA(3S)-GalNAc(6S)	77:23	1.23
CS-13	GlcA(3S)-GalNAc	100	1.00
CS-14	GlcA(2,3S)-GalNAc	100	2.00
CS-15	GlcA(2S)-GalNAc	100	1.00

[a] CS-6 has been provided by Creative Biomart.

**Table 3 polymers-13-00313-t003:** Thermodynamic data of marine CSs-FGF-2 interactions.

Polysaccharide	*k*_D1_ (M)	*k*_D2_ (M)
CS-1^[a]^	1.8 × 10^−5^	8.6 × 10^−8^
CS-2	7.4 × 10^−6^	2.1 × 10^−8^
CS-4	3.4 × 10^−6^	6.8 × 10^−9^

[a] Titration curves are showed as example in Appendix A.

**Table 4 polymers-13-00313-t004:** Thermodynamic data of semi-synthetic CSs-FGF-2 interactions.

Entry	Polysaccharide	*k*_D1_ (M)	*k*_D2_ (M)
**1**	CS-6	1.3 × 10^−4^	1.08 × 10^−6^
**2**	CS-7^[a]^	1.31 × 10^−6^	3.88 × 10^−9^
**3**	CS-15^[a]^	3.6 × 10^−6^	7.77 × 10^−9^
**4**	CS-13	6.6 × 10^−6^	-
**5**	CS-14	8.6 × 10^−6^	3.37 × 10^−8^
**6**	CS-12	4.0 × 10^−8^	2.9 × 10^−6^
**7**	CS-8	1.75 × 10^−8^	2.56 × 10^−6^
**8**	CS-11	1.33 × 10^−6^	4.75 × 10^−9^

[a] Titration curves are showed as examples in Appendix A.

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
