# Peer review of "Deciphering Structural Determinants in Chondroitin Sulfate Binding to FGF-2: Paving the Way to Enhanced Predictability of Their Biological Functions"

_polymers, 2021, doi:10.3390/polym13020313_

Round 1

Reviewer 1 Report

The authors report here the analysis of the interactions between FGF-2, a model heparin-binding growth factor, and a set of chondroitin sulfate (CS) polysaccharides including natural marine and semi-synthetic polysaccharides with different compositions. The polymers were first characterized using NMR, circular dichroism, zeta potential measurements, etc, and then the binding with FGF-2 was studied by SPR. Overall, the paper is well-written and the experimental part is complete and correct. However, in order to support publication in Polymers, the manuscript should be revised:

  • Page 6, line 197: please indicate the polysaccharide concentration employed in the zeta potential measurements.
  • Page 8, line 260: the authors determine the ratio between GalNAc (6S,4S) and GalNAc units in CS-7 integrating the CH2 signals of position 6. What about the CH signals of position 4? These signals should be considered in order to rule out the presence of GalNAc 6S (and even GalNAc 4S) units.
  • Page 9, table 2: the sulfation degree value for CS-11 should be higher than 2.
  • Page 9, figure 7a. Surprisingly, although the heparin sulfation degree is much higher, the zeta potential of heparin is much lower (less negative value, -29.5 mV) than those corresponding to the natural CSs (from -31.9 to -41.7mV). How can you explain this result? For comparison purposes, it would be very convenient to include previous literature data on the zeta potentials (Z) of heparin and CS polysaccharides with similar composition/sulfation degrees.
  • Figures 7a and 8a: In order to support (or not) some of the conclusions extracted from the Z values, it would be very useful to show the standard deviations/errors of the zeta potential measurements.
  • Page 10, line 310: “Effectively, when comparing the obtained values, enrichment in 6S resulted in a clear decrease in the zeta potential values, with the exception of CS-4.” This sentence should be corrected. This reviewer cannot see a clear trend between Z and 6S proportion. In fact, CS-4, the only polysaccharide that clearly displays a low 6S percentage, affords the lowest Z value, exactly the opposite trend.
  • Tables 3 and 4: kinetic or thermodynamic data? The authors are showing the thermodynamic dissociation constants. The method to calculate these constants should be better explained. Are they calculated by analysing the whole SPR (association, dissociation) curves? In this case, the kinetic association and dissociation constants are also obtained and should be provided. Or are the dissociation constants obtained by considering the SPR values at the equilibrium state? In this second case, the SPR values at the steady state are usuallly plotted against the polysaccharide concentration and the fitting of this curve provides the thermodynamic dissociation constant.

Page 12, line 397: The interaction between CS-E and FGF-2 has been previously characterized for example, in J. Biol. Chem. 277, 43707–43716, 2002. The results obtained for CS-7 could be compared with those reported in previous papers.

Author Response

Answers to Reviewer 1

The authors report here the analysis of the interactions between FGF-2, a model heparin-binding growth factor, and a set of chondroitin sulfate (CS) polysaccharides including natural marine and semi-synthetic polysaccharides with different compositions. The polymers were first characterized using NMR, circular dichroism, zeta potential measurements, etc, and then the binding with FGF-2 was studied by SPR. Overall, the paper is well-written and the experimental part is complete and correct.

We thank the reviewer 1 for his/her very nice comments on our manuscript.

Reviewer’s 1st comment: Page 6, line 197: please indicate the polysaccharide concentration employed in the zeta potential measurements.

Author’s reply: We thank reviewer 1 for pointing out this mistake. The polysaccharide concentration employed in the zeta potential measurements has been added in page 6, line 196.

Reviewer’s 2nd comment: Page 8, line 260: the authors determine the ratio between GalNAc (6S,4S) and GalNAc units in CS-7 integrating the CH2 signals of position 6. What about the CH signals of position 4? These signals should be considered in order to rule out the presence of GalNAc 6S (and even GalNAc 4S) units.

Author’s reply: The CH signal of unsulfated position 4 was not considered for the integration since it is generally smaller volume than the CH2 signals and it is usually very close to other densities in the DEPT-HSQC spectrum (see for example Figure 6b,c,d). These make the integration difficult and the obtained values are poorly reliable.

Another point to be considered in the sulfation pattern estimation of CS-7 is the protection pattern of the corresponding synthetic intermediate, polysaccharide 1. As reported in our previous work [ref. 24: Vessella G. et al. Mar. Drugs 2019, 17(12), 655–670], derivative 1 has about 90% of repeating units displaying the free O-4,6 diol and the remaining 10% still carrying a benzylidene ring protection, in accordance with the adopted semi-synthetic strategy. It is highly likely that the unsulfated GalNAc units derive from the benzylidene protected repeating units on 1, while other sulfation patterns are unreliable.

Reviewer’s 3rd comment: Page 9, table 2: the sulfation degree value for CS-11 should be higher than 2.

Author’s reply: The sulfation degree value for CS-11 has been corrected in table 2 (page 9) and in figure 8 (page 11).

Reviewer’s 4th comment: Page 9, figure 7a. Surprisingly, although the heparin sulfation degree is much higher, the zeta potential of heparin is much lower (less negative value, -29.5 mV) than those corresponding to the natural CSs (from -31.9 to -41.7mV). How can you explain this result? For comparison purposes, it would be very convenient to include previous literature data on the zeta potentials (Z) of heparin and CS polysaccharides with similar composition/sulfation degrees.

Author’s reply: We thank reviewer 1 for pointing out these results. Certainly, heparin possess a higher sulfation degree than natural CSs. However, it is important to note that zeta potential is not a measure of the total charge of the polymer, being a physicochemical parameter related to the surface charge of the polysaccharide. By this reason, in zeta potential values, has an importantly effect the 3D-structure of the polysaccharide and, particularly, the arrangement of sulfates on the surface. Taking into consideration this, polysaccharides with higher sulfation degrees, such as heparin, can have higher values of zeta potential (less negative value) than others with lower sulfation degrees, such as the natural CSs employed in this work. Finally, we would like to emphasize that similar behaviors have been observed previously by us and other authors when different CSs [ref 15: Benito-Arenas, R. et al. Carbohydr. Polym. 2018, 202, 211-218] and other sulfated polysaccharides [ref 37: Doncel-Pérez, E. et al. Carbohydr. Polym. 2018, 191, 225-233; ref 38: Revuelta, J. et al. ACS Appl. Mater. Interfaces 2020, 12, 25534−25545 and Yuan, L. et al. Colloids and Surfaces B, 2009, 73, 346-350] were analysed.

In respect to the last question, whilst zeta potential is frequently employed to analyse polysaccharide-protein interactions [for recent examples see: Comert, F. et al. Phys. Chem. Chem. Phys. 2017, 19, 21090; Bertini, S. et al. Clin. Appl. Thromb Hemost. 2017, 23(7), 725-734; Sommers, C.D. et al. J. Pharm. Biomed. Anal. 2017, 5, 113-121]; however, results on the sulfated polysaccharides zeta potentials are scarce and are not related to each other, which makes difficult their comparison.

In fact, to the best of our knowledge, the first systematic study to analyse the effect of location and degree of sulfation in chondroitin sulfate surface net charge (zeta potential) is which described by us recently [ref 15: Benito-Arenas, R. et al. Carbohydr. Polym. 2018, 202, 211-218].  For comparison purposes, we have included a table in Supporting Information with Z-potential values of several polysaccharides with similar composition (Table S6).

Reviewer’s 5th comment: Figures 7a and 8a: In order to support (or not) some of the conclusions extracted from the Z values, it would be very useful to show the standard deviations/errors of the zeta potential measurements.

Author’s reply: We thank reviewer 1 by this suggestion. We have included the standard errors in both figures.

Reviewer’s 6th comment: Page 10, line 310: “Effectively, when comparing the obtained values, enrichment in 6S resulted in a clear decrease in the zeta potential values, with the exception of CS-4.” This sentence should be corrected. This reviewer cannot see a clear trend between Z and 6S proportion. In fact, CS-4, the only polysaccharide that clearly displays a low 6S percentage, affords the lowest Z value, exactly the opposite trend.

Author’s reply: We thank reviewer 1 for pointing out these results. During the manuscript revision, we have detected a mistake in the zeta potential values that we have corrected. In particular, zeta potential values of CS-3 and CS-5 were exchanged. We agree with referee that considering the zeta potential values for marine polysaccharides, a trend between Z and 6S proportion is not clear. In this regard, this sentence has been revised. Nevertheless, we would like to emphasize that the binding capacity of polysaccharides could be explained by their zeta potential values.

Reviewer’s 7th comment: Tables 3 and 4: kinetic or thermodynamic data? The authors are showing the thermodynamic dissociation constants. The method to calculate these constants should be better explained. Are they calculated by analysing the whole SPR (association, dissociation) curves? In this case, the kinetic association and dissociation constants are also obtained and should be provided. Or are the dissociation constants obtained by considering the SPR values at the equilibrium state? In this second case, the SPR values at the steady state are usuallly plotted against the polysaccharide concentration and the fitting of this curve provides the thermodynamic dissociation constant.

Author’s reply: We thank reviewer 1 for pointing out the mistake. Effectively, tables 3 and 4 show thermodynamic data and not kinetic data. The constants have been obtained by considering the SPR values at the equilibrium state. We have introduced an example in Fig. S5 of Supporting Information.

Reviewer’s 8th comment: Page 12, line 397: The interaction between CS-E and FGF-2 has been previously characterized for example, in J. Biol. Chem. 277, 43707–43716, 2002. The results obtained for CS-7 could be compared with those reported in previous papers.

Author’s reply: We thank reviewer 1 by this suggestion. Certainly, previous papers have characterized the interaction between CS-E and FGF-2. However, we have not considered the direct comparison of our results with previously ones because CS-E derivatives that have been used to study CS-E and FGF-2 interactions are CS-E- enriched squid cartilage, which is a mixture of different disaccharides with over 50% E-units, or chemically synthetized CS-E oligosaccharides, what makes the results hardly comparable. It is worth noting that some studies using chondroitin sulfate derivatives have shown an important dependence of binding affinity on sulfation degree and profile. Despite this, and according with reviewer 1 suggestion, we have included a sentence in the manuscript about these previous studies in which the relevance of 3D-helical structure  in binding affinity to FGF-2 is explained.

Reviewer 2 Report

In the manuscript “Deciphering Structural Determinants in Chondroitin Sulfate Binding to FGF-2: Paving the Way to Enhanced Predictability of their Biological Functions”, the characterization of several polysaccharides of marine origin and a library of semi-synthetic polysaccharides and their binding affinity with FGF-2 were investigated. However, this manuscript needs minor repair. The comments and problems are as follows:

  1. Abstract: The authors over stated the purpose of the research and stated briefly the contents of the research and the principal results. I would prefer including more numerical data, if possible.

  1. In line 112, please refer to a reference to extract CSs.

  1. In line 131, what is a known polysaccharide (1)?

  1. In lines 138-139, why the authors adjusted the pH to 12, and then added HCl until neutralization? Why the PH of the solution was set to 12 instead of being slightly alkalinity?

  1. Figure 7b, horizontal axis, it should be wavelength.

  1. In line 403, why the author selected CS-7 and CS-15?

  1. In figure 8a, please employ heparine as reference. And figure 8b, the differences of CS6-15 were not clear, suggest use different colors to distinguish them.

Author Response

Answers to Reviewer 2

In the manuscript “Deciphering Structural Determinants in Chondroitin Sulfate Binding to FGF-2: Paving the Way to Enhanced Predictability of their Biological Functions”, the characterization of several polysaccharides of marine origin and a library of semi-synthetic polysaccharides and their binding affinity with FGF-2 were investigated. However, this manuscript needs minor repair. The comments and problems are as follows:

Reviewer’s 1st comment: Abstract: The authors over stated the purpose of the research and stated briefly the contents of the research and the principal results. I would prefer including more numerical data, if possible.

Author’s reply: We thank reviewer 2 for this suggestion. We have included some numerical data in the abstract.

Reviewer’s 2nd comment: In line 112, please refer to a reference to extract CSs.

Author’s reply: We thank reviewer 2 for this suggestion. We have removed the text, referring the extraction processes to  references 19-22. [ref 19: Blanco, M. Mar. Drugs 2015, 13, 3287-3308; ref 20: Vázquez, J.A.et al. Food Chem. 2016, 198, 28–35; ref 21: Vázquez, J.A. et al. Mar. Drugs 2018, 16, 344-359 and ref 22: Vázquez, J.A.

Carbohydr. Polym. 2019, 210, 302-313.

Reviewer’s 3rd comment: In line 131, what is a known polysaccharide (1)?

Author’s reply: We thank reviewer 2 for this question. Polysaccharide 1 is a semi-synthetic partially protected chondroitin polysaccharide prepared according to a reported procedure, as cited in the manuscript [ref. 24: Vessella, G.; et al. Mar. Drugs 2019, 17(12), 655–670]. Polysaccharide 1 is obtained from unsulfated microbial-sourced chondroitin sodium salt in five steps. These include: (1) treatment on cation-exchange resin to convert the carboxylate moieties into its acid form; (2) methyl esterification step, repeated twice (AcCl, CH3OH, rt, overnight, overall degree of substitution 0.92); (3) regioselective protection at the O-4,6 diol on GalNAc units with a benzylidene ring (a,a-dimethoxytoluene, CSA, DMF, 80 °C, overnight, degree of substitution 0.95); (4) acetylation of the O-2,3 diol on GlcA units (Ac2O, Et3N, DMAP, CH3CN, rt, overnight degree of substitution 2); (5) acid hydrolysis of the benzylidene group on GalNAc units (90% v/v aq. AcOH, 50 °C, 48 h, degree of restored diol 0.87). Here below, other references where the procedure from step 1 to 4 was employed for chemical regioselective modification of microbial-sourced chondroitin: ref. 11 in the manuscript [Bedini, E. et al. Angew. Chem. Int. Ed. 2011, 50, 6160-6163]; ref. 25 in the manuscript [Laezza, A. et al. Chem. - Eur. J. 2016, 22(50), 18215–18226].

Reviewer’s 4th comment: In lines 138-139, why the authors adjusted the pH to 12, and then added HCl until neutralization? Why the pH of the solution was set to 12 instead of being slightly alkalinity?

Author’s reply: The pH value of 12 is necessary in order to accomplish a quantitative alkaline hydrolysis of all the acetyl groups and methyl ester moieties, installed on the polysaccharide chain and still present after sulfation and acid hydrolysis steps. The alkaline hydrolysis is performed overnight at room temperature, then the mixture is neutralized to quench the reaction and to ensure a neutral pH during the subsequent dialysis step. It is important to underline that, despite the alkalinity of the reaction mixture, no polysaccharide degradation due to an a,b-elimination mechanism on GlcA unit was detected as confirmed by the absence of typical signals around 6.0 ppm in the 1H NMR spectrum of CS-7.

Reviewer’s 5th comment: Figure 7b, horizontal axis, it should be wavelength.

Author’s reply: We thank reviewer 2 for pointing out the mistake. We have added horizontal axis legend (wavelength / nm).  

Reviewer’s 6th comment: In line 403, why the author selected CS-7 and CS-15?

Author’s reply: We thank reviewer 2 for this question. Chondroitin sulfate derivatives CS-7 and CS-15 have been selected in base to two factors:

  • Both produce a similar responsiveness against FGF-2 (kD1=31 x 10-6, kD2=3.88 x 10-9 and kD1=3.6 x 10-6, kD2=7.77 x 10-9 for CS-7 and CS-15 respectively).
  • Both have completely different 3D-structures. Thus, a 3D-helical structure has been assigned to CS-15 in base to the minimum observed at 210 nm in its CD spectrum (see Figure 8b), whilst CS-7 seems to be an un-structured polysaccharide in base to the absence of signal in its CD spectrum.

To test the 3D-structure preference of growth factor-chondroitin sulfate interactions, we think to perform SPR competition experiments to see the impact of those structure on the interactions. Effectively, the SPR competition experiments with both CS-derivatives clearly showed growth factor-CS interaction is impacted by the 3D-polysaccharide structure, which seems to determinate the binding zone.

Reviewer’s 7th comment: In figure 8a, please employ heparine as reference. And figure 8b, the differences of CS6-15 were not clear, suggest use different colors to distinguish them.

Author’s reply: We thank reviewer 2 by these suggestions. We have included heparine in Figure 8a as reference and colors have been employed in Figure 8b to differ CD spectrum of polysaccharides CS-6CS-15.

Round 2

Reviewer 1 Report

The authors have carefully revised the paper and properly answered to all the questions. The manuscript have been improved and therefore I recommend now publication in Polymers in the present form. 

Minor point: 

  • In the abstract, please check the affinity values: 1.31-130 micromolar for the first step (not nanomolar); 3.88 nM-1.08 uM for the second one.